# α-Synuclein oligomers form by secondary nucleation

Catherine K. Xu [1,2], Georg Meisl [1], Ewa A. Andrzejewska [1], Georg Krainer [1,3], Alexander J. Dear[1,4], Marta Castellana-Cruz[1], Soma Turi[1], Irina A. Edu [1], Giorgio Vivacqua[5,6], Raphaël P. B. Jacquat [1], William E. Arter[1], Maria Grazia Spillantini[6], Michele Vendruscolo [1], Sara Linse [4] & Tuomas P. J. Knowles [1,7] ✉

Oligomeric species arising during the aggregation of α-synuclein are implicated as a major source of toxicity in Parkinson's disease, and thus a major potential drug target. However, both their mechanism of formation and role in aggregation are largely unresolved. Here we show that, at physiological pH and in the absence of lipid membranes, α-synuclein aggregates form by secondary nucleation, rather than simple primary nucleation, and that this process is enhanced by agitation. Moreover, using a combination of single molecule and bulk level techniques, we identify secondary nucleation on the surfaces of existing fibrils, rather than formation directly from monomers, as the dominant source of oligomers. Our results highlight secondary nucleation as not only the key source of oligomers, but also the main mechanism of aggregate formation, and show that these processes take place under conditions which recapitulate the neutral pH and ionic strength of the cytosol.

The process of protein aggregation is associated with over 50 human disorders, including Alzheimer's and Parkinson's diseases (PD)[1]. In PD, aggregates of the 14 kDa protein α-synuclein are the major component of Lewy bodies and neurites, which have emerged as the pathological hallmarks of the disease[2]. In addition to its abundance in the characteristic amyloid deposits in PD, α-synuclein is further implicated as a causal agent in PD disease development by the finding that duplications and triplications of the WT α-synuclein gene, as well as a number of single-point mutations, are associated with familial cases of PD[3–5].

While deposits of fibrillar protein are hallmarks of protein aggregation diseases, oligomeric intermediates are as implicated as the major source of toxicity, as the high molecular weight fibrillar aggregates are typically relatively inert in a biological context[1,2,6–8]. Moreover, determining oligomer dynamics is crucial for the elucidation of aggregation mechanisms[9,10]. Oligomers are nevertheless relatively poorly characterized, due to challenges in their analyses that render them invisible to most conventional biophysical techniques, namely their low abundance, transient nature, and high degree of heterogeneity[8–15].

In the case of the Alzheimer's disease-associated Aβ peptide, oligomer dynamics have been used to determine the molecular pathways of fibril formation[16]. For α-synuclein, several studies of oligomer dynamics have employed single molecule FRET experiments, which have observed the interconversion of oligomers with different FRET efficiencies[11,17–19]. However, such measurements to date have not uncovered the source of oligomers as the aggregation reaction progresses[20].

Here, we exploit the power of single molecule microfluidics to study the dynamics of α-synuclein oligomers under native-like conditions[21]. Combined with bulk assays and chemical kinetics, this approach allows us to quantitatively determine the molecular

[1]Centre for Misfolding Diseases, Yusuf Hamied Department of Chemistry, University of Cambridge, Cambridge, UK. [2]Max Planck Institute for the Science of Light, Erlangen, Germany. [3]Institute of Molecular Biosciences (IMB), University of Graz, Graz, Austria. [4]Biochemistry and Structural Biology, Lund University, Lund, Sweden. [5]Integrated Research Center (PRAAB), Campus Biomedico University of Rome, Rome, Italy. [6]Department of Clinical Neurosciences, University of Cambridge, Cambridge, UK. [7]Cavendish Laboratory, University of Cambridge, Cambridge, UK. ✉e-mail: tpjk2@cam.ac.uk

mechanism of $\alpha$-synuclein aggregation under conditions that reflect the neutral pH and ionic strength of the cytosol (Fig. 1).

## Results

### $\alpha$-Synuclein aggregation occurs via secondary pathways

Although $\alpha$-synuclein oligomers are implicated as toxic species in PD, the molecular mechanisms by which both they and high molecular weight aggregates form remain largely unknown, despite substantial efforts. However, in order to enable the rational design of drugs that target aggregation, for example, inhibiting the formation of toxic oligomeric species, this mechanistic information is required. Detailed investigations into the individual microscopic processes involved have often only been possible under reaction conditions that differ from cytosolic pH and/or ionic strength and often required the investigation of different mechanistic steps at disparate conditions; primary nucleation induced by synthetic lipids, secondary nucleation at moderate ionic strength/acidic pH or high ionic strength/neutral pH, and elongation at moderate ionic strength/neutral pH[22–26]. From such studies, we now qualitatively understand certain aspects of the aggregation mechanism but have so far not achieved a quantitative description that can account for the experimental data. In order to unify the reaction steps into one complete mechanistic description of fibril formation from $\alpha$-synuclein, we established experimental conditions for the reproducible aggregation of $\alpha$-synuclein at neutral pH (Fig. 2 and Supplementary Figs. S6, S7, and S8)[27].

We investigated the fibril formation kinetics of $\alpha$-synuclein in the absence and presence of varying concentrations of fibrillar seeds. Upon the addition of fibrillar seeds, the lag phase and aggregation half-time ($t_{1/2}$) decreased in a seed concentration-dependent manner (Fig. 2). This behaviour is highly characteristic of the presence of secondary processes, whereby existing fibrils catalyse the formation of further fibrils[28,29]. By contrast, in the absence of secondary processes, the addition of seed fibrils would not significantly alter the aggregation kinetics. Moreover, fitting our data globally to kinetic models, we found that the aggregation kinetics are inconsistent with a mechanism which does not include secondary processes; $\alpha$-synuclein therefore aggregates via secondary processes[30] (Supplementary Table S1).

### Secondary nucleation is the dominant mechanism of fibril formation

The dependence of aggregation kinetics on protein concentration can be used to infer the molecular mechanisms of fibril formation, given a careful consideration of the underlying reaction steps[28,30]. In our case,

the dependence of the unseeded aggregation rates on the total monomer concentration gives rise to a scaling exponent of −0.5, the proportionality factor in the relationship between the logarithms of the aggregation half-times and monomer concentrations. This concentration dependence is consistent with fibril formation mechanisms of both fragmentation and saturated secondary nucleation[24,28]. In the former case, the number of fibrils increases by the fragmentation of existing fibrils. In the latter case of saturated secondary nucleation, monomers quickly bind to fibril surfaces, and their subsequent conversion to aggregates and release into solution, which is independent of the free monomer concentration, is the rate limiting step[24,28].

In order to determine which of the two mechanisms is dominant, the fibril lengths in the plateau region can be measured to estimate the fragmentation rate. In the plateau region, the monomer concentration, and therefore the rate of secondary nucleation is minimal, however, fragmentation continues at the same speed as during the aggregation reaction. Therefore, changes in the length distribution in the plateau phase can be used to estimate the fragmentation rate directly. We obtained the fibril length distributions by transmission electron microscopy (TEM) imaging of aggregation mixtures throughout the plateau phase, finding a decrease in the mean fibril length over time (Fig. 3, and Supplementary Figs. S11, and S12). To incorporate our length distribution-derived fragmentation rates into our model of $\alpha$-synuclein aggregation here, samples were withdrawn from aggregation mixtures under the exact same conditions as the fitted kinetic data in Fig. 2. From fitting an analytical expression for the average length (for derivation see "Methods") to the fibril lengths measured by TEM (Fig. 3), we obtain an upper bound on the rate of fibril accumulation due to fragmentation $\kappa(\text{frag}) = 0.01 \text{ h}^{-1}$. However, fitting the aggregation kinetics (Fig. 2) yields a very clear result: $\kappa = 0.4 \text{ h}^{-1}$, i.e., 40-fold faster than would be expected based on fragmentation alone. Similarly, fitting the aggregation kinetics using the rate of aggregate formation due to fragmentation alone completely fails to account for the observed aggregation kinetics (Fig. 3c). Therefore, our fibril length-based analysis indicates that secondary nucleation, not fragmentation, is the main mechanism of fibril formation.

However, measurements of fibril length distributions may not always be fully representative of the true distribution. Surface-based measurements such as TEM may be limited by non-equal capture of fibrils of different lengths, while solution-based methods such as dynamic light scattering (DLS) are often biased towards larger objects and assume spherical geometry. We thus employed a complementary approach to establishing the dominance of secondary nucleation over

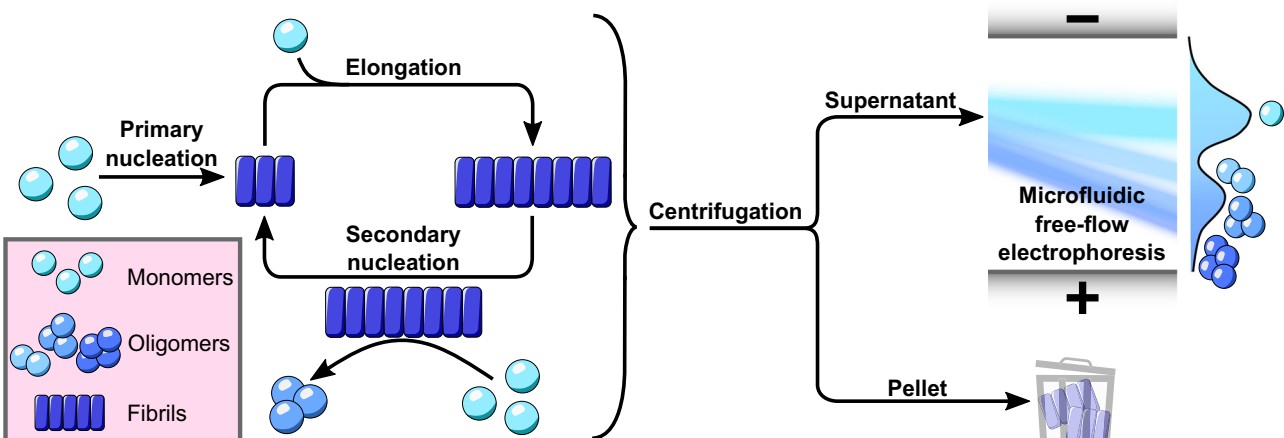

**Fig. 1 | Using microfluidic free-flow electrophoresis with single molecule detection, we were able to fractionate $\alpha$-synuclein aggregation mixtures and determine oligomer dynamics.** By using chemical kinetics we determined that secondary nucleation on fibril surfaces is the dominant mechanism of both $\alpha$-synuclein oligomers and new fibrils.

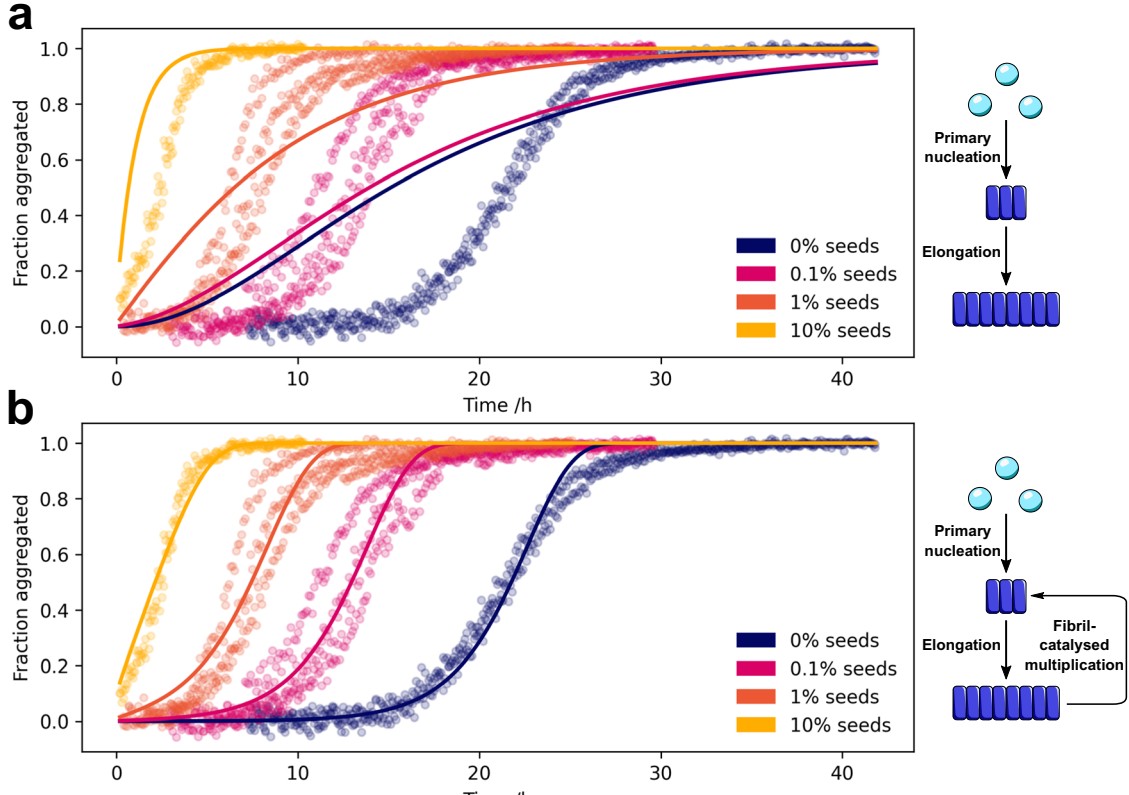

**Fig. 2 | Secondary processes are involved in the aggregation of α-synuclein.** The aggregation kinetics of AlexaFluor-488-labelled α-synuclein were followed by aggregation-induced quenching of the AlexaFluor-488 dye (Fig. S7). The aggregation kinetics of the labeled and unlabelled α-synuclein are the same (Fig. S6 and Table S1). The dependence of α-synuclein aggregation kinetics on the absence and presence of varying concentrations of fibrillar seeds. Data were fitted to kinetic models in the absence (**a**) and presence (**b**) of secondary processes. The data are only consistent with a model that includes the fibril-catalyzed formation of new fibrils.

fragmentation. The Brichos domain, a molecular chaperone, is a well-established inhibitor of secondary nucleation in amyloid aggregation of both the Alzheimer's-associated Aβ peptide and, more recently, α-synuclein[31–33]. We confirmed that Brichos inhibits α-synuclein aggregation under our experimental conditions (Supplementary Fig. S13). Since Brichos inhibits secondary nucleation and does not affect fragmentation, we therefore conclude that fragmentation is only a minor contributor to the rate of fibril formation, and that secondary nucleation is the dominant mechanism of α-synuclein fibril formation.

## Oligomers form by secondary nucleation on fibril surfaces

In order to elucidate further details of the secondary nucleation process, we investigated α-synuclein oligomer dynamics during aggregation. We previously demonstrated the ability of μFFE to fractionate complex aggregation mixtures and resolve oligomeric subpopulations according to their electrophoretic mobilities, a function of radius and charge (Fig. S14)[21]. We have further demonstrated its extension to single molecule spectroscopy to maximize information on fractionated species[34]. Here, we employ μFFE at the single molecule level to monitor oligomer mass concentrations during α-synuclein aggregation to yield insights into oligomer dynamics.

A key feature of this approach in studying oligomers is its minimal perturbation of the reaction system. The sample under study is rapidly diluted and fractionated in solution just a few milliseconds prior to measurement, a timescale on which the sample composition is unlikely to change. This contrasts with more traditional single molecule approaches requiring the almost million-fold dilution of samples, or size exclusion chromatography, where samples interact differentially with a solid matrix on a timescale of minutes to hours[11,16,20]. Moreover,

the method is agnostic to oligomer structure, as oligomers are detected by their intrinsic fluorescent label, in contrast to antibody-based methods such as ELISA[35–38]. The reaction mixture is therefore minimally perturbed for the measurement; the attachment of the AlexaFluor-488 fluorophore at position 122 did not affect the aggregation kinetics compared to the WT protein (Fig. S6 and Supplementary Table S1), likely due to its location outside of the fibril core[39,40]. Due to the high concentration of monomer and thus background noise, we used a simple photon count thresholding approach to estimate the oligomer concentration. However, the non-uniformity of the confocal spot laser intensity means that the estimated oligomer concentrations may be on the order of ten- or even thousand-fold smaller than the true concentrations (see SI for details).

Using our microfluidic approach, we observed the maximum in oligomer mass concentration slightly before the half-time. Crucially, by seeding the reaction, this peak in oligomer mass concentration was shifted in time and again similarly located close to the half-time (Fig. 4). This oligomer dependence on fibril seeds indicates that oligomers form predominantly via a fibril-catalysed mechanism, rather than directly from monomers. If oligomers formed directly from monomers, then their formation rate should be affected only by the monomer concentration and the introduction of seeds would not be expected to simply result in a shift in time of the otherwise unchanged oligomer peak, as observed in Fig. 4. These observations therefore point to secondary nucleation as the dominant mechanism of oligomer formation (Supplementary Fig. S15).

To further verify this mechanistic conclusion, we derived the rate laws describing aggregation and oligomer formation (see "Methods" for details of the kinetic model) and fitted the measured oligomer

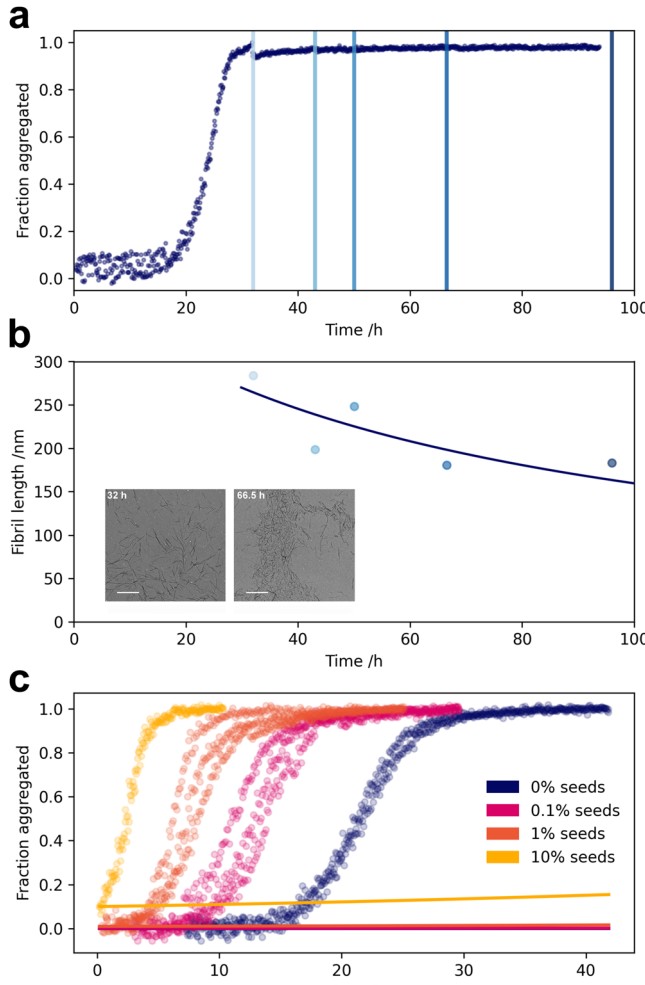

**Fig. 3 | Fragmentation does not account for α-synuclein fibril proliferation.** Fibrils were withdrawn from an aggregation reaction at the indicated timepoints in the plateau phase of the aggregation mixture (**a**) and imaged by TEM (**b**), inset, scale bar = 1 μM). **b** The mean lengths of fibrils were determined from at least 8 TEM images containing a minimum of 650 fibrils in total, and fitted to kinetic models to determine the fragmentation rate, finding a very low value of 0.01 h$^{-1}$. **c** The kinetic data were then fitted with fragmentation as the mechanism of fibril amplification, with the fragmentation rate constant fixed to the value determined in (**b**).

concentrations using the rate constants for fibril formation determined above (Supplementary Table S2). This model fits the data well and allows determination of bounds on the rate constants of oligomer formation and dissociation: oligomers are formed at a rate greater than $4 \times 10^{-5}\,\text{s}^{-1}$ per mole of fibrils at a monomer concentration of 100 μM. By comparison, Aβ secondary oligomers are formed at a rate of $3 \times 10^{-5}\,\text{s}^{-1}$ at a monomer concentration of 5 μM[16]. Additionally, the oligomer dissociation rate was fast on the timescale of the aggregation reaction, i.e., hours or faster, as shown by the fact that oligomer concentration decline closely tracks monomer consumption during the aggregation reaction. Such oligomers are therefore only detectable due to the fast timescale of our microfluidic approach. In summary, secondary nucleation is therefore not only responsible for the formation of new fibrils, but is also the main source of oligomers.

### Oligomers form under quiescent, native-like conditions
We next investigated the role of shaking in α-synuclein aggregation. Under fully quiescent conditions, aggregation proceeded at a much lower rate, demonstrating that shaking increases the rate of aggregation. However, the addition of seeds drastically reduced the $t_{1/2}$,

indicating that secondary processes still take place in the absence of agitation. In order to determine which microscopic processes/es are catalyzed by shaking, we studied the oligomer content of quiescent, seeded aggregation reactions using μFFE at the single molecule level before and after moderate shaking (10 min, 200 rpm). Prior to shaking, very few oligomers were observed, but the concentration of oligomers increased by more than a factor of three following moderate shaking (Fig. 5). The post-shaking concentration was also higher than the oligomer concentration during the aggregation plateau phase under constant shaking at the same speed (200 rpm), demonstrating that the majority of these oligomers did not arise through fragmentation of fibrils induced by the shaking. Given that the conversion and dissociation steps are rate-limiting in this system, these must be the processes affected by agitation. From a mechanical perspective, the dissociation of oligomers from fibrils is more likely to be catalyzed by shaking. We therefore conclude that α-synuclein oligomers form by secondary nucleation under quiescent conditions at neutral pH, and that shaking accelerates their formation, likely by facilitating their dissociation from fibril surfaces.

### Parkinson's disease patient samples catalyse oligomer formation through secondary nucleation
Having characterised the in vitro aggregation mechanism in detail, we next explored its links to Parkinson's disease. Due to the low concentration of aggregates in CSF[41,42], we chose quiescent aggregation conditions to minimize the primary nucleation rate, in order to maximize the observable seeding effect. While the aggregation of wells seeded with pooled CSF from healthy subjects appeared to be stochastic, the CSF from Parkinson's disease patients consistently induced α-synuclein aggregation (Figs. 6 and Supplementary Fig. S16). Given that CSF has been found to inhibit aggregation[43], these data clearly indicate that aggregates in the disease CSF are sufficient to overcome this inhibition and seed α-synuclein aggregation. We additionally performed RT-QuIC analyses of brain homogenates from patients with several synucleinopathies, including PD, and our labelled α-synuclein (Supplementary Fig. S17), demonstrating that patient brain-derived aggregates are able to seed our labelled α-synuclein, thus supporting its use as a model system for the investigation of molecular mechanisms in disease. Moreover, using our microfluidic platform, we investigated the oligomer content of the Parkinson's CSF-seeded aggregation reaction, detecting oligomers with the same biophysical properties, namely size and electrophoretic mobility, as our in vitro-generated oligomers (Fig. 6). These data thus provide evidence for secondary nucleation as an oligomer production mechanism in Parkinson's disease.

### Discussion
In this study, we have demonstrated that α-synuclein oligomers form by secondary nucleation under physiologically relevant conditions; namely neutral pH and a salt concentration mimicking the osmotic pressure in the cytosol. These oligomers are also the dominant mechanism of the formation of new fibrils, allowing an exponential increase in aggregate mass concentration over time. Previous work has suggested that the contribution of secondary processes to α-synuclein aggregation is highly pH-dependent[22,24,44]. The presence of secondary processes at neutral pH was hinted at by data from Buell et al., and with our detailed investigation, we have determined that this process is secondary nucleation and not fragmentation[22]. Moreover, several additional studies have reported seed-concentration dependent aggregation kinetics at neutral pH, suggesting that secondary nucleation is a general feature of α-synuclein aggregation[44–48].

Similarly, previous studies on α-synuclein oligomer dynamics have generally focused on the role of primary nucleation in oligomer formation[11,18,19,49]. In light of our findings herein, a re-examination of these data shows that they are in fact consistent with a peak in

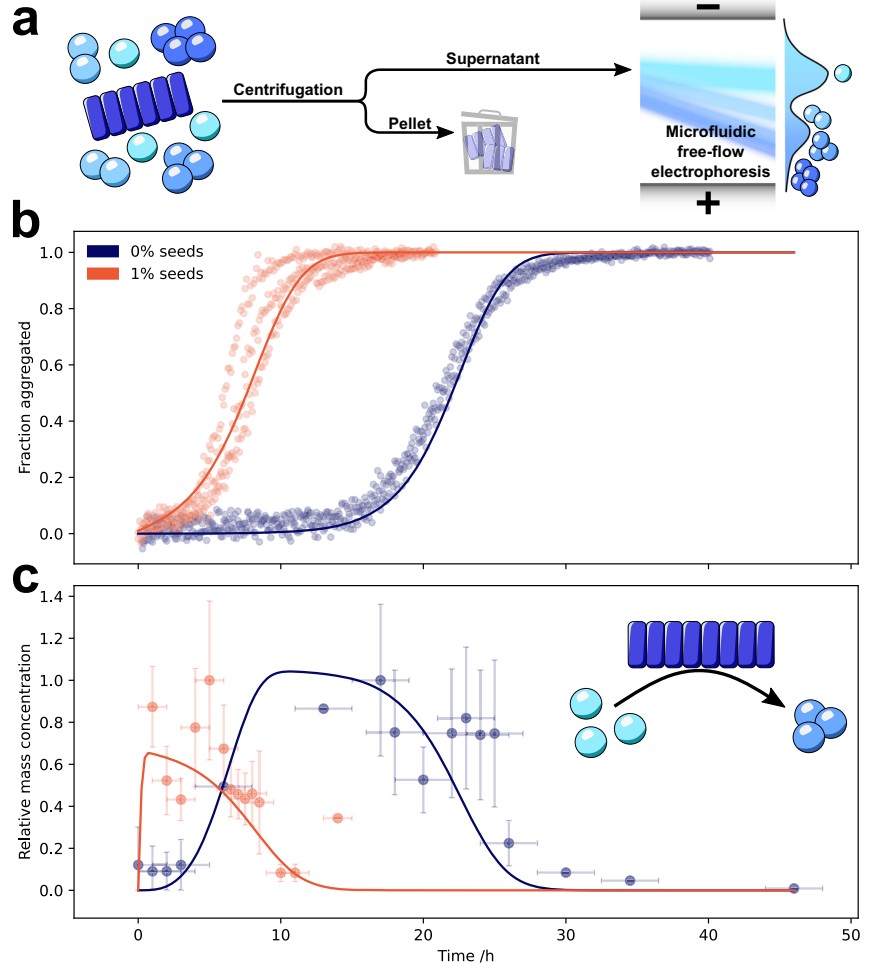

**Fig. 4 | α-Synuclein oligomers form by secondary nucleation.** Aggregation mixtures at various time points throughout the reaction were centrifuged (21, 130 × g, 10 min, 20 °C) to remove large fibrillar aggregates and the oligomer content of the resulting supernatant studied by μFFE at the single molecule level (**a**). Kinetics of fibrillar α-synuclein in unseeded (blue) and seeded (red, 1% seeds) aggregation reactions, measured by fluorescence quenching, are shown alongside the fitted model (**b**). The relative oligomer mass concentrations were determined (**c**) and fitted to a model in which oligomers can form via both primary nucleation from monomers and secondary nucleation on fibril surfaces (model details in SI). *X*-axis error bars represent the time range over which data were averaged, which corresponds to the standard deviation of the aggregation half times. Where present, *y*-axis error bars represent the standard error of the mean oligomer mass concentrations from up to 8 biological replicates; points without *y*-axis error bars represent a single sample.

concentration close to the aggregation half-time that we observe here, and thereby with a secondary nucleation formation mechanism. By investigating both fibril and oligomer dynamics in detail, we quantitatively elucidate the molecular mechanism of α-synuclein aggregation. This was made possible by our development of both aggregation conditions under cytosolic pH and ionic strength, and an oligomer detection method which is both blind to oligomer structure and minimally perturbs the reaction mixture. Through this study, we established that, while agitation is known to be able to induce fibril fragmentation and primary nucleation, under these conditions it markedly increases the rate of secondary nucleation[50–52]. Primary nucleation is likely a heterogeneous process, where agitation is believed to accelerate its rate by increasing turnover at the catalytic surface[53]. Secondary nucleation is similarly a heterogeneous process, the only difference being the catalytic surface is fibrils, therefore it stands to reason that a similar accelerating effect can also affect this process[54]. This is analogous to crystallisation processes, for which mild agitation can increase the rate of secondary nucleation by facilitating detachment, a finding which forms the basis of the use of mild agitation in industrial crystallisation processes[55–58].

In conclusion, we identify secondary nucleation as the dominant process for the formation of both oligomers and fibrils in α-synuclein aggregation. α-Synuclein oligomers therefore provide both a potential source of toxicity and mechanism of aggregate spreading in PD, which is not only limited to early stages of the disease, given that oligomer formation is catalysed by fibrillar aggregates[8,59]. The detailed mechanistic framework we have elucidated can thus be used to better understand the role of α-synuclein in PD pathology and how to effectively develop therapeutic strategies. Our finding that secondary nucleation is not only a pathway to the formation of new fibrils but also the main source of oligomeric species highlights it as a promising therapeutic target with dual effectiveness: stopping secondary nucleation will stop oligomer production in the short term and slow fibril accumulation in the long term.

## Methods
### Ethical Statement
Brain samples from four patients were used in this paper for RT-QuIC analysis, consisting of 3 male (healthy control, PD, and MSA) and 1 female (DLB). The were provided by the UK Brain bank and the

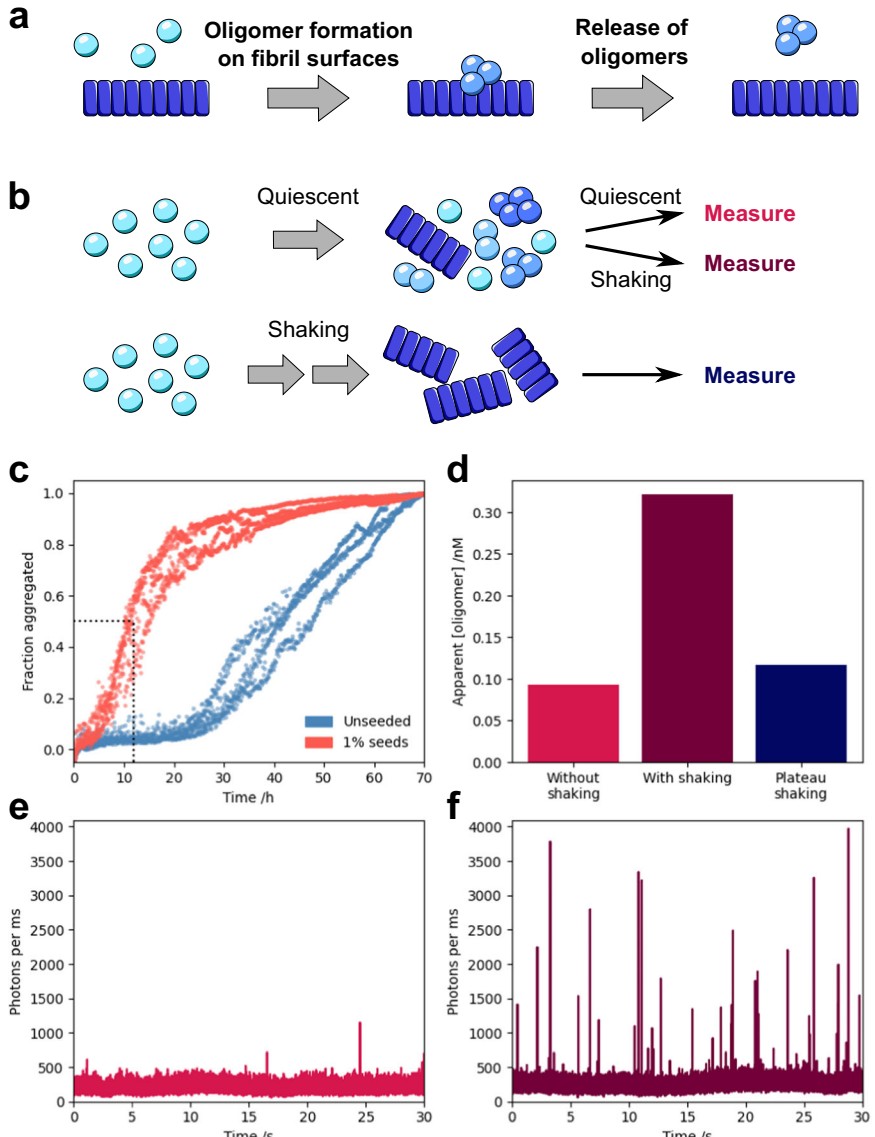

**Fig. 5 | Secondary nucleation proceeds under quiescent, physiological conditions.** Secondary nucleation involves the formation of oligomers on fibril surfaces, followed by their release into solution (**a**). α-Synuclein was aggregated under both quiescent and shaking conditions (**b**, **c**), and the oligomer populations investigated. The relative oligomer mass concentrations of quiescent seeded (1%) aggregation reactions at the half-time (indicated by the dotted line in (**c**) before and after shaking for 10 minutes at 200 rpm are shown alongside the oligomer concentration during the plateau phase of the reaction under shaking (**d**). Example timetraces of photon count rates are shown for quiescent seeded (1%) aggregation reactions before (**e**) and after (**f**) shaking.

ethical approval was obtained from the Cambridgeshire 2 Research Ethics Committee.

## Purification of α-synuclein

α-Synuclein (WT or N122C variant) was overexpressed in *Escherichia coli* BL21 cells. The cells were centrifuged (20 min, 3985 × g, 4 °C; JLA-8.1000 rotor, Beckman Avanti J25 centrifuge (Beckman Coulter)), and the pellet resuspended in buffer (10 mM tris, 1 mM EDTA, protease inhibitor) prior to lysis by sonication on ice. Debris was removed by centrifugation (20 min, 39,121 × g, 4 °C; JLA-25.5 rotor), and the supernatant incubated (20 min, 95 °C). Heat-sensitive proteins were removed by centrifugation (15 min, 39,121 × g, 4 °C; JLA-25.5 rotor). Subsequent incubation with streptomycin sulfate (10 mg/mL, 15 min, 4 °C) precipitated out DNA. α-Synuclein was extracted from the supernatant (15 min, 39,121 × g, 4 °C; JLA-25.5 rotor) by the gradual addition of ammonium sulfate (361 mg/mL). The α-synuclein-containing pellet was collected by centrifugation (15 min, 39,121 × g, 4 °C; JLA-25.5 rotor) and

resuspended in buffer (25 mM tris, pH 7.4). Dialysis was used for complete buffer exchange, and the resulting mixture run on a HiLoad™ 26/10 Q Sepharose high performance column (GE Healthcare), at room temperature. Under a gradient of 0–1.5 M NaCl over 600 mL, α-Synuclein was eluted at ~350 mM. Selected fractions were fractionated at room temperature on a Superdex 75 26/600 (GE Healthcare) and eluted in PBS (pH 7.4). For the N122C variant, 3 mM dithiothreitol (DTT) was added to all buffers to prevent dimerization. The concentration of α-synuclein was determined by absorbance at 275 nm, using a molar extinction coefficient of 5600 $M^{-1}$ $cm^{-1}$. Aliquots were then flash-frozen in liquid nitrogen and stored at −80 °C.

## Labelling of α-synuclein

The N122C variant of α-synuclein was fluorescently labeled with AlexaFluor-488 dye. DTT was removed from purified α-synuclein by buffer exchange into PBS using P10 desalting columns containing Sephadex G25 matrix (GE Healthcare). The DTT-free protein was

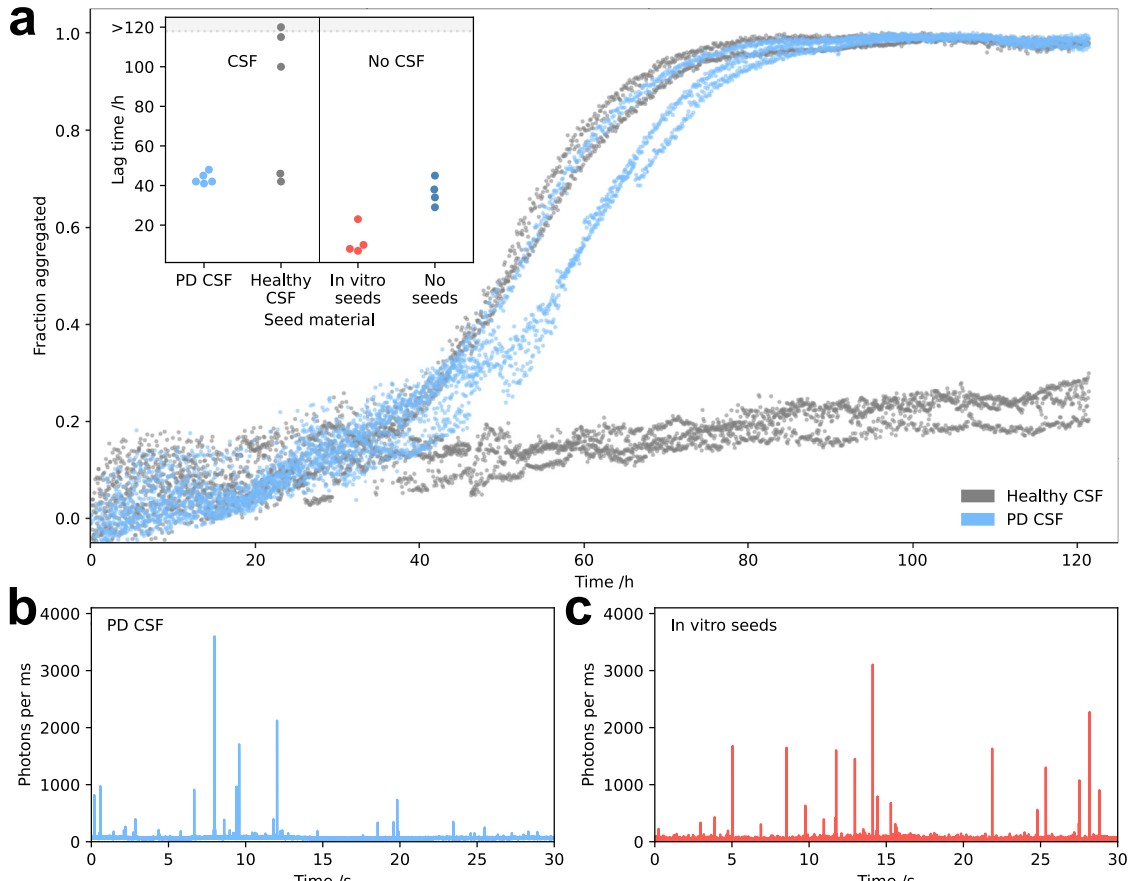

**Fig. 6 | CSF from Parkinson's disease patients catalyses oligomer formation.**
**a** Aggregation kinetics of α-synuclein aggregation in the presence of 4% v/v pooled CSF from Parkinson's disease patients and a healthy cohort. The extracted lag times (time taken to reach 25% aggregation) are shown alongside unseeded and seeded (1% mass concentration) reactions in the absence of CSF (inset). **b**, **c** Example photon count timetraces from aggregation reactions seeded by PD CSF (**b**) and in vitro-generated seeds (**c**). Oligomers from the PD CSF-seeded reaction were investigated by μFFE at around 30% aggregation (**b**) and found to have similar biophysical properties to a corresponding sample from the in vitro seeded reaction (**c**).

incubated (overnight, 4 °C, rolling system) with a 1.5x molar excess of AlexaFluor-488 dye functionalized with a maleimide moiety. Excess unbound dye and α-synuclein dimers were removed by eluting the mixture over P10 desalting columns containing Sephadex G25 matrix (GE Healthcare). The resulting α-synuclein concentration was estimated by the dye absorbance at 495 nm, using a molar extinction coefficient of 72,000 M$^{-1}$ cm$^{-1}$. Aliquots were flash-frozen in liquid nitrogen and stored at -80 °C for up to 3 weeks prior to experiments.

## Aggregation of α-synuclein

Aggregation of α-synuclein was carried out in non-binding 96-well plates (Corning) at 37 °C in a FLUOstar Omega microplate reader (BMG Labtech). Each well contained 100 μL of reaction mixture and a glass bead. The buffer used was Dulbecco's PBS (pH 7.4) with 0.01% (w/v) sodium azide. Interwell areas and empty wells were filled with PBS prior to sealing the plate with a foil cover. For experiments under shaking conditions, plates were shaken for 355 s at 200 rpm between each reading cycle; quiescent reactions were read at the same rate, but in the absence of all shaking. WT reactions were followed by the addition of 50 μM thioflavin T, whereas labelled N122C reactions (100% labelled protein) were monitored by AlexaFluor-488 fluorescence (Supplementary Fig. S7). Fibrils from unseeded reactions of 100 μM monomer under the same buffer and shaking conditions were used directly as seeds for seeding reactions, with no sonication (Supplementary Figs. S9 and Supplementary Fig. S10). For CSF-seeded aggregation reactions, CSF was added to a total of 4% volume in

100 μM α-synuclein under the same quiescent conditions as described above, and the sample subjected to shaking (10 min, 200 rpm) prior to measurement by FFE. CSF biospecimens used in the analyses presented in this article were obtained from The Michael J. Fox Foundation for Parkinson's Research.

## RT-QuIC

Brain samples from UK brain bank and with confirmed autoptic diagnosis of synucleinopathies were employed in the present study. The samples included: Parkinson's Disease (PD) cingulate cortex, Multiple System Atrophy (MSA) occipital cortex, Dementia with Lewy bodies (DLB) occipital cortex and healthy control occipital cortex. Brain homogenates (BH; 10% w/v) were prepared by homogenizing the tissue in 40 mM PBS containing 1% protease inhibitor (Halt, Thermo Scientific - 1860932) and 0.1% phenylmethylsulfonyl fluoride (PMSF), using a Bead Beater (Biospec Products; 11079110z) for 2 minutes at maximum speed. The homogenate was then spun at 3000 × $g$ for 5 minutes at 4 °C and the supernatant was transferred to a new 0.5 ml lowBind Eppendorf tube and stored at −80 °C until RT-QuIC analysis. RT-QuIC reactions were performed in black 96-well plates with a clear bottom (Nalgene Nunc International). Plates were preloaded with 3 silica beads (1 mm diameter, BioSpec Products) per well. For BH-seeded reactions, 4 μL of the indicated BH was added to wells containing 95 μL of the reaction buffer to give final concentrations of 40 mM phosphate buffer (pH 8.0), 170 mM NaCl, 100 μM of monomeric AlexaFluor-488-labelled N122C α-synuclein (filtered through a 100 kDa

MWCO filter immediately prior to use). The plate was then sealed with a plate sealer film (Nalgene Nunc Inter- national) and incubated at 37.5 °C in a BMG FLUOstar Omega plate reader with cycles of 1 min shaking (500 rpm double orbital) and 15 minutes rest throughout the indicated incubation time. Fluorescence measurements (490 ± 5-nm excitation and 520 ± 5-nm emission; bottom read) were taken every 15 min. Each sample was run in three technical replicates.

## Analysis of bulk kinetic data

The signal obtained during aggregation kinetics (ThT fluorescence or AlexaFluor-488 fluorescence) was taken to be proportional to the fibril mass present in the sample. The data were then fitted using the AmyloFit Platform and following the protocol in Meisl et al. to a model including primary nucleation, elongation and secondary nucleation with reaction order 0[30]. This model was able to describe the data well across concentrations. To produce misfits, the same data were fitted with a model including only primary nucleation and elongation.

## Measurement of fibril length distributions

At certain timepoints in the plateau phase of the aggregation reaction, 1 $\mu$L of the reaction was withdrawn and the plate returned to the plate-reader. The reaction sample was mixed with 9 $\mu$L PBS (pH 7.4), and applied to a transmission electron microscopy (TEM) grid (continuous carbon film on 300 mesh Cu). Following adsorption, the sample was washed with milliQ water ($2 \times 10 \mu$L). Samples were negatively stained with uranyl acetate (2% w/v, 10 $\mu$L, 2 min) and washed with milliQ water ($2 \times 10 \mu$L). TEM grids were glow discharged using a Quorum Technologies GloQube instrument at a current of 25mA for 60s. TEM images were obtained using a Thermo Scientific (FEI Company, Hillsboro, OR) Talos F200X G2 microscope operated at 200 kV. TEM images were recorded using a Ceta 4k × 4k CMOS camera. The lengths of imaged fibrils were manually determined with ImageJ (example images in Figure S9). The lengths of between 650 and 1550 individual fibrils were measured for each sample.

## Analysis of fibril length distributions

The fibril length in the plateau phase of the reaction, during which the fibril mass concentration is constant, can be modeled by the following equations. By definition, in the plateau phase, the aggregate mass concentration is constant, i.e., $M(t) = M_\infty$. While nucleation processes become negligible when the monomer is depleted, fragmentation still takes place, thus the number concentration of fibrils, $P(t)$, is given by

$$\frac{dP}{dt} = k_- M_\infty, \tag{1}$$

which can be solved to yield

$$P(t) = k_- M_\infty t + P_{plateau}, \tag{2}$$

where $k_-$ is the fragmentation rate constant, $t$ is the time since the plateau was first attained and $P_{plateau}$ is the number concentration of fibrils at $t = 0$.

The mean length, $L(t)$, at the plateau is thus given by:

$$L(t) = \frac{M(t)}{P(t)} = \frac{M_\infty}{k_- M_\infty t + P_{plateau}} = \frac{1}{k_- t + \frac{1}{L_{plateau}}}, \tag{3}$$

where $L_{plateau}$ is the average length when the plateau is first attained. Using the steady-state expression for the average length during an aggregation reaction derived e.g., in Cohen et al.[60–62] as an estimate for $L_{plateau}$, the rate of fibril formation due to fragmentation is approximately given by $\kappa(frag) = L_{plateau} k_- = 0.01\,h^{-1}$. This is thus an estimate of the rate of fibril formation based purely on measurements of fibril lengths which can then be compared with kinetic measurements of the

actual rate of fibril accumulation $\kappa$ to see if this is consistent with a purely fragmentation-driven mechanism.

## Fabrication of microfluidic free-flow electrophoresis devices

Microfluidic free-flow electrophoresis ($\mu$FFE) devices were designed and fabricated as follows. Briefly, acetate masks were used to produce SU-8 molds of devices by photolithography, the heights of which were measured with a profilometer (Dektak, Bruker, Billerica, MA). Poly-dimethylsiloxane (PDMS; 1:10 mixture of primer and base, Dow Corning) was applied to the mold and baked (65 °C, 1.5 h). $\mu$FFE devices were then excised and biopsy punches used to create holes for tubing and electrode connections, with diameters of 0.75 mm and 1.5 mm, respectively. Following sonication in isopropanol (5 min), devices were bonded to glass coverslips (#1.5) by activation with oxygen plasma. Immediately prior to use, prolonged oxygen plasma treatment was used to hydrophilize device surfaces.

## $\mu$FFE device operation

The $\mu$FFE device design used contains liquid electrodes (3 M KCl solution containing 1 nM Atto-488 dye) to connect the electrophoresis chamber to the external electric circuit[21,63]. These liquid electrodes were connected to the circuit via hollow metal electrodes made from bent syringe tips, which also constituted the outlets for the liquid electrodes. Samples were flowed into the device at controlled flow rates by the use of syringe pumps (Cetoni neMESYS, Korbussen, Germany), connected to polytetrafluoroethylene (PTFE) tubing (0.012" inner diameter × 0.030" outer diameter, Cole-Parmer, St. Neots, UK). The flow rates used were 1000, 200, 140, and 10 $\mu$L h$^{-1}$ for the auxiliary buffer (15× diluted PBS in milliQ water), electrolyte, desalting milliQ water, and sample, respectively. The electric field was applied by a benchtop power supply (Elektro-Automatik EA-PS 9500-06, Viersen, Germany) connected to the metal electrode outlets.

## Acquisition of $\mu$FFE data

$\mu$FFE data were acquired using laser confocal fluorescence microscopy; a 488 nm wavelength laser beam (Cobolt 06-MLD 488 nm 200 mW diode laser, Cobolt, Stockholm, Sweden) was coupled into single-mode optical fibre (P3-488PM-FC01, Thorlabs, Newton, NJ) and collimated (60FC-L-4-M100S-26, Schäfter und Kirchhoff, Hamburg, Germany) before being directed into the back aperture of an inverted microscope body (Applied Scientific Instrumentation Imaging, Eugene, OR). The laser beam was then reflected by a dichroic mirror (Di03-R488/561, Semrock, Rochester, NY) and focused to a concentric diffraction-limited spot in the microfluidic channel through a high-numerical-aperture water-immersion objective (CFI Plan Apochromat WT 60x, NA 1.2, Nikon, Tokyo, Japan). Photons arising thorugh fluorescence emission were detected using the same objective. Fluorescence was then passed through the dichroic mirror and imaged onto a 30 $\mu$m pinhole (Thorlabs), removing out of focus light. The signal was then filtered through a bandpass filter (FF01-520/35-25, Semrock), and focused onto a single-photon counting avalanche diode (APD, SPCM-14, PerkinElmer Optoelectronics, Waltham, MA). Photons were recorded using a time-correlated single photon counting (TCSPC) module (TimeHarp 260 PICO, PicoQuant, Berlin, Germany) with 25 ps time resolution. Single-photon counting recordings were obtained using custom-written Python code.

Aggregation samples (100 $\mu$L) were withdrawn from the plate at various times during the aggregation reaction and centrifuged (21,130 × g, 10 min, 20 °C). The top 70 $\mu$L was carefully withdrawn without disturbing the pellet containing large aggregates. An aliquot of the supernatant was then diluted in PBS to ~5 $\mu$M total monomer mass concentration and injected into the device. An electric field of 300 V was applied and photon count timetraces obtained at 5–10 positions laterally distributed across the field direction for a total of at least 1 min per position.

## Analysis of μFFE data

A detailed account of the analysis of μFFE data is provided in the SI.

## Fitting of oligomer dynamics

The coarse-grained rate equations governing oligomers (concentration $S(t)$) formed by primary nucleation during an amyloid fibril formation reaction are:

$$\frac{dS}{dt} = k_{o1}m(t)^{n_{o1}} - k_{e1}S(t), \qquad (4)$$

where $m(t)$ is the concentration of monomeric protein, $k_{o1}$ is the rate constant for oligomer formation from primary nucleation with reaction order $n_{o1}$, and $k_{e1}$ is the rate constant for dissociation of oligomers to monomers. In a system dominated by secondary nucleation of oligomers, the rate equations are instead well-approximated by:

$$\frac{dS}{dt} = k_{o2}m(t)^{n_{o2}}M(t) - k_{e2}S(t)M(t), \qquad (5)$$

where $M(t)$ is the mass concentration of amyloid fibrils and $k_{o2}$ is the rate constant for oligomer formation from fibril-dependent processes, with reaction order $n_{o2}$. For simplicity, we modelled $M(t)$ using the analytical expressions given in ref. 51 with rate parameters chosen as the values determined by fitting the bulk data on fibril formation in the main text (Fig. 2 and Table S2). Eqs (4) and (5) were then fitted numerically to the experimental data on oligomer concentration using python. Eq. (5) provided the superior fit, supporting the conclusion that oligomers are formed predominantly by secondary processes in this assay.

## Reporting summary

Further information on research design is available in the Nature Portfolio Reporting Summary linked to this article.

## Data availability

The data generated in this study have been deposited in the Zenodo database under the accession code https://doi.org/10.5281/zenodo.11958462. Source data are provided in this paper.

## Code availability

Analysis code is available at https://github.com/cx220/aS_kinetics[64].

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

## Acknowledgements

We thank Dr Manuela R Zimmermann and Minghao Zhang for helpful discussions on data analysis. We additionally thank Dr Heather Greer for her help with the acquisition of TEM images and the EPSRC Underpinning Multi-User Equipment Call (EP/P030467/1) for funding the TEM. We would like to acknowledge funding from the European Research Council under the European Union's Horizon 2020 research and innovation program through the ERC grant DiProPhys (agreement ID 101001615), and the following sources: Herchel Smith Research Studentship (C.K.X.), Herchel Smith Fellowship (G.K.), Wolfson College Junior Research Fellowship (G.K.), Marie Skłodowska-Curie grant MicroSPARK (agreement no. 841466; G.K.), Swedish Research Council (VR 2015-00143; S.L.), The Addenbrooke's Charity Trust (M.G.S., G.V.), Parkinson's UK (M.G.S.).

## Author contributions

C.K.X., G.K., W.E.A., M.V., S.L., and T.P.J.K. conceived the study. C.K.X. and M.C.C. developed the α-synuclein kinetics assay. C.K.X., E.A.A., and I.A.E. performed the kinetics experiments. C.K.X., E.A.A., and G.K. acquired µFFE data. G.V and M.G.S. performed RT-QuIC experiments. C.K.X., G.M., and S.T. developed a theory for data analysis. C.K.X., G.M., A.J.D., R.P.B.J., and W.E.A. contributed software. C.K.X., G.M., and A.J.D. analyzed data. C.K.X. and G.M. wrote the manuscript with input from all authors.

## Competing interests

At the time of initial submission, Georg Meisl and Alexander J Dear were employees of Wavebreak Therapeutics (formerly Wren Therapeutics). Michele Vendruscolo, Sara Linse, and Tuomas PJ Knowles are cofounders of Wavebreak Therapeutics (formerly Wren Therapeutics). The remaining Authors declare no competing interests.
