## [Peer Review File · Nature Communications]

Reviewers' Comments:

Reviewer #1:

Remarks to the Author:

The manuscript is a carefully written and executed study from a well-respected group of authors.

The kinetics of α -synuclein aggregation has been studied for more than a decade and the dominant role of secondary nucleation has been reported

See for example:

Ref 23 - Gaspar, R. et al. Secondary nucleation of monomers on fibril surface dominates α -synuclein aggregation and provides autocatalytic amyloid amplification. Quarterly Reviews of Biophysics 50 (2017)

and also at physiological pH (add to reference list)

ACS Chem Neurosci 2023 Sep 6;14(17):3125-3131. Secondary Processes Dominate the Quiescent, Spontaneous Aggregation of α -Synuclein at Physiological pH with Sodium Salts Robert I Horne Georg Meisl Michele Vendruscolo

A strength in this study lies in the use of microfluidic free-flow electrophoresis (μ FFE) to monitor fibril formation. Recently reported: Arter, W. E. et al. Rapid structural, kinetic, and immunochemical analysis of α -synuclein oligomers in solution. Nano Letters 20, 8163–8169 (2020).

Using this approach a more complete picture of α -synuclein fibril formation kinetics has been achieved.

Minor Comments

- 1) FIGURE 2 is completely missing from the draft!
- 2) Line 85- need a reference to justify /explain scaling exponent and saturated secondary nucleation as the readership may not be familiar
- 3) Page 5 line 98- need to explicitly mention conditions for fragmentation measurements here
- 4) Figure 3b, fragmentation is minimal are fibrils close to steady state, so hard to measure fragmentation rate? This needs a little discussion.
- 5) Line 138-139, needs a bit more discussion to justify this
- 6) Fig 4c- model is not a great fit to experimental data, is the experimental data really able to show the model is appropriate?
- 7) The idea that agitation can impact secondary nucleation more than fragmentation is an important point, and this is not perhaps always appreciated. Has this been reported elsewhere? if so need to add references.

Reviewer #2:

Remarks to the Author:

The manuscript by Xu et al from the Linse/Knowles groups entitled "Rapid single molecule detection reveals that α -synuclein oligomers form by secondary nucleation" investigates the role of secondary nucleation during α -synuclein aggregation with particular focus on oligomer formation. With a sound single molecule detection approach aggregation-time dependent oligomer formation (with artificially tagged synuclein) could be identified quantitatively. This work on the one hand consolidates a lot of known qualitative observations demonstrating the kinetic role of secondary nucleation in great details. In addition, it is observed that the almost omnipresent shaking during aggregation is key for oligomer formation attributed to be dissolved from the fibril surface. In short, the data presented are very important for the amyloid aggregation field. There are a few remaining questions:

- (i) Figure 2 is missing, it is thus difficult to assess the quality of the work entirely.

(ii) It is stated that the seeds used are from fibrils without further treatment (i.e. sonication). What is the quality of the seeds in terms of fibril length? What happens if the fibrils will be sonicated? Is the secondary nucleation then less effective in respect to fibril elongation (i.e. more relative amount of ends versus fibrils surface). It is suggested that it is noted that the fibril seeds used here, differ from the ones usually used by the Linse/Knowles group (where often low pH seeds were used).

(iii) The fibril length studied by TEM could be not correct because small seeds may not be seen, or may not be on the grid. Could the authors not use their microfluidics device to get an additional estimate on fibril distribution length or by another method?

Reviewer #3:

Remarks to the Author:

The group has been exploring aggregation pathway of alpha-synuclein over the years. Herein the authors present a thorough and detailed analysis of the kinetics of a-Syn aggregation by micro fluidic free-flow electrophoresis at the single molecule level.

Secondary nucleation has been proposed and characterized by the same group (although at different conditions) in previous works (PMID: 37701726; PMID: 37578897; PMID: 36939645; PMID: 36802433). In addition, micro fluidic free-flow electrophoresis has been employed to resolve oligomeric species of aSyn and other amyloidogenic proteins (PMID: 33079553; PMID: 36746944). Based on these observations, the manuscript as it is, although interesting and technically sound, does not bring enough novelty to justify publication in such a prestigious journal.

I recommend including PD brain material as seed (from patients), to allow elucidation of a more realistic situation. One must bear in mind that fibril polymorphism exists, and the seed characteristic may change the aggregation kinetic. There is a recent publication that explores this issue (<https://pubmed.ncbi.nlm.nih.gov/36802500/>).

TEM is not a quantitative technique; thus, it is not straightforward to comment on fibril fragmentation based solely on the TEM images.

Authors comment that they use physiologically relevant conditions, but the assay medium is far from physiological. One must consider mimicking the crowded cellular environment.

It would be valuable to discuss how to address PD therapeutically based on the different aggregation processes (primary or secondary nucleation).

Reviewer #4:

Remarks to the Author:

Review of:

Rapid single molecule detection reveals that α -synuclein oligomers form by secondary nucleation
Xu et al.

Nice paper!

1. Key results; Clarity and Context; Significance

In the abstract, the authors wrote:

"Here we first show that, at physiological pH, α -synuclein aggregates by secondary nucleation, rather than fragmentation, and that this process is enhanced by agitation."

Conclusions of this reviewer:

- The perception (importance of secondary nucleation) is not new.
- The use of single molecule detection is not a key, compelling new technology applied in such investigations and offering new insights. New variants of ultra-resolution microscopy, as one counterexample, could/would have been. In any event, one might wish to (have) emphasize(d) the methodology, perhaps in a submission to something like Nature Methods (?), along the lines "Rapid single molecule detection in microfluidic free-flow electrophoresis confirms that α -synuclein oligomers form by secondary nucleation".

The authors do not cite a number of recent publications (examples: refs. r1-r13 below; the first 8 published in 2023) dealing with important aspects of the secondary nucleation of AS aggregation, i. e. key factors determining the course of reaction: modification status of the AS monomer, solution conditions, temperature, agitation, small molecule reaction inhibitors or accelerators. The uncited references listed below include 7 papers (r1, r4, r5, r7, r8, r9, r12) co-authored by one or both of the senior authors of the present submission, M. Vendruscolo and T.P.J. Knowles. In fact, the titles (and abstracts) of r5 and the present submission offer the same observation regarding secondary nucleation. In addition, the authors have pre-published virtually the identical paper (bioRxiv), which apparently did not lead to significant changes, improvements. [This reviewer confesses to being a non-proponent of prepublication.]

r1. Staats, R., Brotzakis, Z.F., Chia, S., Horne, R.I., and Vendruscolo, M. (2023). Optimization of a small molecule inhibitor of secondary nucleation in alpha-synuclein aggregation. *Front Mol Biosci* 10, 1155753. 10.3389/fmolb.2023.1155753.

r2. Pandit, E., Das, L., Das, A.K., Dolui, S., Saha, S., Pal, U., Mondal, A., Chowdhury, J., Biswas, S.C., and Maiti, N.C. (2023). Single point mutations at the S129 residue of alpha-synuclein and their effect on structure, aggregation, and neurotoxicity. *Front Chem* 11, 1145877. 10.3389/fchem.2023.1145877.

r3. Majid, N., Siddiqi, M.K., Hassan, M.N., Malik, S., Khan, S., and Khan, R.H. (2023). Inhibition of primary and secondary nucleation along with disruption of amyloid fibrils and alleviation of associated cytotoxicity: A biophysical insight of a novel property of Chlorpropamide (an anti-diabetic drug). *Biomater Adv* 151, 213450. 10.1016/j.bioadv.2023.213450.

r4. Horne, R.I., Murtada, M.H., Huo, D., Brotzakis, Z.F., Gregory, R.C., Possenti, A., Chia, S., and Vendruscolo, M. (2023). Exploration and Exploitation Approaches Based on Generative Machine Learning to Identify Potent Small Molecule Inhibitors of alpha-Synuclein Secondary Nucleation. *J Chem Theory Comput* 19, 4701-4710. 10.1021/acs.jctc.2c01303.

r5. Horne, R.I., Metrick, M.A., 2nd, Man, W., Rinauro, D.J., Brotzakis, Z.F., Chia, S., Meisl, G., and Vendruscolo, M. (2023). Secondary Processes Dominate the Quiescent, Spontaneous Aggregation of alpha-Synuclein at Physiological pH with Sodium Salts. *ACS Chem Neurosci* 14, 3125-3131. 10.1021/acscchemneuro.3c00282.

r6. Frey, L., Ghosh, D., Qureshi, B.M., Rhyner, D., Guerrero-Ferreira, R., Pokharna, A., Kwiatkowski, W., Serdiuk, T., Picotti, P., Riek, R., and Greenwald, J. (2023). On the pH-dependence of α -synuclein amyloid polymorphism and the role of secondary nucleation in seed-based amyloid propagation. *bioRxiv* p. 10.1101/2023.06.25.546428.

r7. Dada, S.T., Hardenberg, M.C., Toprakcioglu, Z., Mrugalla, L.K., Cali, M.P., McKeon, M.O., Klimont, E., Michaels, T.C.T., Knowles, T.P.J., and Vendruscolo, M. (2023). Spontaneous nucleation and fast aggregate-dependent proliferation of alpha-synuclein aggregates within liquid condensates at neutral pH. *Proc Natl Acad Sci U S A* 120, e2208792120. 10.1073/pnas.2208792120.

r8. Bell, R., Castellana-Cruz, M., Nene, A., Thrush, R.J., Xu, C.K., Kumita, J.R., and Vendruscolo, M. (2023). Effects of N-terminal Acetylation on the Aggregation of Disease-related alpha-synuclein Variants. *J Mol Biol* 435, 167825. 10.1016/j.jmb.2022.167825.

r9. Zurlo, E., Kumar, P., Meisl, G., Dear, A.J., Mondal, D., Claessens, M., Knowles, T.P.J., and Huber, M. (2021). In situ kinetic measurements of alpha-synuclein aggregation reveal large population of short-lived oligomers. *PLoS One* 16, e0245548. 10.1371/journal.pone.0245548.

r10. Sakunthala, A., Datta, D., Navalkar, A., Gadhe, L., Kadu, P., Patel, K., Mehra, S., Kumar, R., Chatterjee, D., Sengupta, K., et al. (2021). Size-dependent secondary nucleation and amplification of α -synuclein amyloid fibrils. *bioRxiv* 10.1101/2021.12.28.474324.

r11. Peduzzo, A., Linse, S., and Buell, A.K. (2020). The Properties of alpha-Synuclein Secondary Nuclei Are Dominated by the Solution Conditions Rather than the Seed Fibril Strain. *ACS Chem Neurosci* 11, 909-918. 10.1021/acscchemneuro.9b00594.

r12. Tornquist, M., Michaels, T.C.T., Sanagavarapu, K., Yang, X., Meisl, G., Cohen, S.I.A., Knowles, T.P.J., and Linse, S. (2018). Secondary nucleation in amyloid formation. *Chem Commun (Camb)* 54, 8667-8684. 10.1039/c8cc02204f.

r13. Roberti, M.J., Folling, J., Celej, M.S., Bossi, M., Jovin, T.M., and Jares-Erijman, E.A. (2012). Imaging nanometer-sized alpha-synuclein aggregates by superresolution fluorescence localization microscopy. *Biophys J* 102, 1598-1607. 10.1016/j.bpj.2012.03.010.

2. Results; Validity; Data and Methodology

- The authors wrote: ". Despite its heavy implication in Parkinson's disease, the aggregation of α -synuclein remains relatively poorly characterized."

Not true, there are few processes in biology that have received more attention.

- α -Synuclein aggregation occurs via secondary pathways. This section is ok

- Secondary nucleation is the dominant mechanism of fibril formation.

There is a problem here, due to the lack of an evaluation/interpretation by the microscopy of Figs. 3, S9, S10, with respect to the incidence and length of secondary initiated fibrils. Such a visualization has been performed in the past, such as by super-resolution fluorescence microscopy (r13 above) in which secondary anchors were visualized and the rate of elongation (nm/h) of secondary fibrils was estimated as "7 \pm 4 nm/min". The authors of the present submission have actually published a mean (ensemble: primary+secondary) elongation rate (Supplementary of the seminal paper Buell et al., 2014: " At 37 ° C and 20 μ M soluble protein concentration, α -synuclein fibrils grow on average 1-2 nm/min in PBS buffer, assuming a fibril diameter of 7 nm, and using the value of k + determined here."). A careful consideration of these values based on comparison of conditions, etc. is beyond the scope of this review, yet considering that secondary fibrils may grow by different rates of radial as well as axial accretion of monomers, the direct visualization and evaluation of primary and secondary fibrils is essential. The methods (Fiji based) used on the TEM data of the present submission did not permit this nor do the authors address the importance of the elongation rate directly.

- fitting of oligomer dynamics

The submitting Cambridge lab has published extensively on kinetic models for amyloid aggregation. Nonetheless, this reviewer chooses to question features of Eqs. 4 and 5 used to evaluate the kinetic data of the present submission. Eq. 4 assumes a concerted (questionable?) interaction of no1 monomers to form a progenitor fibril and also assumes a simple first order dissociation of monomers from fibrils. However, the indications are that monomers can dissociate along the length of a fibril as well as from the ends such that a proper representation would require an additional length dependent term. The same consideration applies to Eq. 5. It is neither the desire nor intention of this reviewer to engage in a protracted "kinetic" controversy but it may be expedient for the authors to reconsider some of the assumptions of their model(s). If some modifications are necessary/indicated, the quantitative data analysis and the conclusions of the paper need to be reexamined.

- Oligomers form by secondary nucleation on fibril surfaces

Th authors applied μ FFE to examine the oligomer dynamics: "A key feature of this approach in studying oligomers is its minimal perturbation of the reaction system. The sample under study is rapidly diluted and fractionated in solution just a few milliseconds prior to measurement, a timescale on which the sample composition is unlikely to change." Fig. S11 shows that oligomers

have a higher mobility than monomers in the electric field. However, the regions selected in the figure seem somewhat arbitrarily placed and the zone between them represents a substantial amount of material. Also why is the monomer peak so broad?

REVIEWER COMMENTS

Reviewer #1 (Remarks to the Author):

The manuscript is a carefully written and executed study from a well-respected group of authors.

The kinetics of α -synuclein aggregation has been studied for more than a decade and the dominant role of secondary nucleation has been reported

See for example:

Ref 23 - Gaspar, R. et al. Secondary nucleation of monomers on fibril surface dominates α -synuclein aggregation and provides autocatalytic amyloid amplification. Quarterly Reviews of Biophysics 50 (2017)

and also at physiological pH (add to reference list)

ACS Chem Neurosci 2023 Sep 6;14(17):3125-3131. Secondary Processes Dominate the Quiescent, Spontaneous Aggregation of α -Synuclein at Physiological pH with Sodium Salts Robert I Horne Georg Meisl Michele Vendruscolo

A strength in this study lies in the use of microfluidic free-flow electrophoresis (μ FFE) to monitor fibril formation. Recently reported: Arter, W. E. et al. Rapid structural, kinetic, and immunochemical analysis of α -synuclein oligomers in solution. Nano Letters 20, 8163–8169 (2020).

Using this approach a more complete picture of α -synuclein fibril formation kinetics has been achieved.

Minor Comments

1) FIGURE 2 is completely missing from the draft!

We apologise for this oversight, figure 2 is present in the resubmitted version.

2) Line 85- need a reference to justify /explain scaling exponent and saturated secondary nucleation as the readership may not be familiar

We thank the reviewer for this point, we have clarified the citations for readers to easily locate the relevant works. We have also edited this section of text to make it easier to follow, with newly inserted text highlighted in bold:

“The dependence of aggregation kinetics on protein concentration can be used to infer the molecular mechanisms of fibril formation, **given a careful consideration of the underlying reaction steps**. The concentration dependence of the unseeded aggregation rates, with a scaling exponent of -0.5, is consistent with fibril formation mechanisms of both fragmentation and saturated secondary nucleation. In the former case, the number of fibrils increases by the fragmentation of existing fibrils. In the latter case **of saturated secondary nucleation**, monomers **quickly** bind to fibril

surfaces, and their subsequent conversion to aggregates **by secondary nucleation** and release into solution is the rate limiting step.”

3) Page 5 line 98- need to explicitly mention conditions for fragmentation measurements here

This is an important point; we have added the following sentence to explicitly include this information: (lines 113-116) “To incorporate our length distribution-derived fragmentation rates into our model of alpha-synuclein aggregation here, samples were withdrawn from aggregation mixtures under the exact same conditions as the fitted kinetic data in Figure 2.”

4) Figure 3b, fragmentation is minimal are fibrils close to steady state, so hard to measure fragmentation rate? This needs a little discussion.

We have now expanded and clarified the explanation in the figure caption as follows (changes highlighted in bold): in the figure caption “Fragmentation is not the dominant mechanism of alpha-synuclein fibril formation. Fibrils were withdrawn from aggregation reactions at the indicated timepoints in the plateau phase of the aggregation mixture (a) and imaged by TEM (b, inset). (b) The mean lengths of fibrils were fitted to kinetic models to determine the fragmentation rate, **finding a very low value of 0.01 h⁻¹**. (c) The kinetic data were then fitted with fragmentation as the mechanism of fibril amplification, **with the fragmentation rate constant fixed to the value determined in (b).**”

Additionally, in the main text we have added some more discussion of this strategy: (lines 105-110) “In order to determine which of the two mechanisms is dominant, the fibril lengths in the plateau region can be measured to estimate the fragmentation rate. In the plateau region, the monomer concentration, and therefore the rate of secondary nucleation is minimal, however, fragmentation continues at the same speed as during the aggregation reaction. Therefore, changes in the length distribution in the plateau phase can be used to estimate the fragmentation rate directly.”

5) Line 138-139, needs a bit more discussion to justify this

We have added a more detailed explanation to explain how this conclusion can be reached even in the absence of fitting detailed mathematical models. With the new text highlighted in bold, this paragraph (lines 168-178) now reads: “Using our microfluidic approach, we observed the maximum in oligomer mass concentration slightly before the half-time. **Crucially, by seeding the reaction, this peak in oligomer mass concentration was shifted in time and again similarly located close to the half-time (Figure 4). This oligomer dependence on fibril seeds indicates that oligomers form predominantly via a fibril-catalysed mechanism, rather than directly from monomers. If oligomers formed directly from monomers, then their formation rate should be affected only by the monomer concentration and the introduction of seeds would**

not be expected to simply result in a shift in time of the otherwise unchanged oligomer peak, as observed in Figure 3. These observations therefore point to secondary nucleation as the dominant mechanism of oligomer formation (Figure S15). “

6) Fig 4c- model is not a great fit to experimental data, is the experimental data really able to show the model is appropriate?

To further improve the ability of our dataset to distinguish between those two mechanisms, we have now taken further measurements of early timepoints during the aggregation reaction, as this is the region where the greatest differences in oligomer dynamics occur between primary nucleation- and secondary nucleation-dominated systems. We have also added misfits, of a model which does not allow for the formation of secondary oligomers in Figure S15, and which cannot account for the data.

7) The idea that agitation can impact secondary nucleation more than fragmentation is an important point, and this is not perhaps always appreciated. Has this been reported elsewhere? if so need to add references.

The acceleration of secondary nucleation processes by agitation has already been employed in industrial crystallisation processes, as noted in the discussion section. To our knowledge, in the context of amyloid aggregation, agitation has only been proposed to promote fragmentation and primary nucleation. We have now included several additional references on this subject and also add the following sentences in the discussion (lines 260-264): “Primary nucleation is likely a heterogeneous process, where agitation is believed to accelerate its rate by increasing turnover at the catalytic surface. Secondary nucleation is similarly a heterogeneous process, the only difference being the catalytic surface is fibrils, therefore it stands to reason that a similar accelerating effect can also affect this process.”

Reviewer #2 (Remarks to the Author):

The manuscript by Xu et al from the Linse/Knowles groups entitled “Rapid single molecule detection reveals that α -synuclein oligomers form by secondary nucleation” investigates the role of secondary nucleation during α -synuclein aggregation with particular focus on oligomer formation. With a sound single molecule detection approach aggregation-time dependent oligomer formation (with artificially tagged synuclein) could be identified quantitatively. This work on the one hand consolidates a lot of known qualitative observations demonstrating the kinetic role of secondary nucleation in great details. In addition, it is observed that the almost omnipresent shaking during aggregation is key for oligomer formation attributed to be dissolved from the fibril surface. In short, the data presented are very important for the amyloid aggregation field. There are a few remaining

questions:

(i) Figure 2 is missing, it is thus difficult to assess the quality of the work entirely.

We apologise for this oversight, figure 2 is present in the resubmitted version.

(ii) It is stated that the seeds used are from fibrils without further treatment (i.e. sonication). What is the quality of the seeds in terms of fibril length? What happens if the fibrils will be sonicated? Is the secondary nucleation then less effective in respect to fibril elongation (i.e. more relative amount of ends versus fibrils surface). It is suggested that it is noted that the fibril seeds used here, differ from the ones usually used by the Linse/Knowles group (where often low pH seeds were used).

In order to minimise potential perturbation of fibril structures between their production and use as seeds, we took fibrils from the endpoints of aggregation reactions under the same conditions as the future seeded reaction, and used them as seeds without additional processing such as sonication or buffer changes. We have tested the effect of sonication (new figure S10) on seeding ability, and found that, as expected, sonicating seeds increases the rate of subsequent seeding reactions.

We have additionally investigated the effect of seed age on seeding capability. We withdrew fibrils for seeding at different timepoints after the aggregation plateau was reached, and found no differences in their seeding ability (new figure S9).

While previous works have employed fibril seeds produced under acidic conditions, since our aggregation reactions are performed under neutral pH, we also formed our fibrils under the same conditions. This was important for two reasons. Firstly, by not changing their buffer medium, we preserved the stability of fibrils. Secondly, we ensured that our seed structures are the same as in our unseeded reactions. Unseeded and seeded data could thereby be globally fitted together as a single dataset, since the fibrils were consistent across all reactions. This global analysis would not be valid if we were to use fibril seeds produced under different conditions (and therefore likely with a different strain structure) from those produced in the unseeded reactions.

(iii) The fibril length studied by TEM could be not correct because small seeds may not be seen, or may not be on the grid. Could the authors not use their microfluidics device to get an additional estimate on fibril distribution length or by another method?

We agree that it is possible for TEM to distort the measured fibril length distributions, due to the deposition step as noted by the reviewer. However, as we are mainly concerned with the changes in length distribution over time, such biases, which will exist in all timepoints, will have a less pronounced effect. Nonetheless, we sought to provide an alternative measure to confirm the dominant mechanism was secondary nucleation not fragmentation. For length distribution measurements, TEM is considered a state-of-the-art method and other surface deposition-based methods such as atomic force microscopy and super-resolution microscopy also come with the same limitations. Solution-based methods such as dynamic light scattering, fluorescence correlation spectroscopy, and microfluidic diffusional sizing, are unsuitable for measuring fibril length distributions given the large size, non-spherical geometry and polydispersity of the aggregation samples.

Therefore, we have also now performed complementary experiments which support our conclusion that fragmentation is not the dominant mechanism by which new fibrils form

which do not rely on measurements of the length distribution. We performed aggregation reactions in the presence of the Brichos chaperone domain, a well-characterised inhibitor of secondary nucleation, but not fragmentation. The aggregation was markedly slowed by brichos, indicating that secondary nucleation, not fragmentation, is the main mechanism of fibril formation. These data are presented in figure S13 and lines 126-139 of the main manuscript text.

Reviewer #3 (Remarks to the Author):

The group has been exploring aggregation pathway of alpha-synuclein over the years. Herein the authors present a thorough and detailed analysis of the kinetics of a-Syn aggregation by micro fluidic free-flow electrophoresis at the single molecule level.

Secondary nucleation has been proposed and characterized by the same group (although at different conditions) in previous works (PMID: 37701726; PMID: 37578897; PMID: 36939645; PMID: 36802433). In addition, micro fluidic free-flow electrophoresis has been employed to resolve oligomeric species of aSyn and other amyloidogenic proteins (PMID: 33079553; PMID: 36746944). Based on these observations, the manuscript as it is, although interesting and technically sound, does not bring enough novelty to justify publication in such a prestigious journal.

We thank the reviewer for this detailed assessment of our work. Secondary nucleation has so far only been reported under acidic (pH 4.8 (Buell et al. PNAS (21), 2014) and 5.5 (Gaspar et al. Q Rev Biophys (50), 2017) conditions, and only for fibril formation. Here, we demonstrate that secondary nucleation not only takes place at neutral pH, but is also the major driver of oligomers, proposed as a key toxic species in Parkinson's disease. This work thus presents a large advance in our understanding of alpha-synuclein aggregation and provides a new consideration of therapeutic design. Previously, secondary nucleation of alpha-synuclein was only thought to be relevant in vivo upon entry into acidic lysosomal compartments. Our new findings demonstrate that secondary nucleation of alpha-synuclein can occur at neutral pH, and even on aggregates extracted from PD brains (new data; see below) and may therefore be a ubiquitous and central process in pathology, contributing both to toxic oligomer production and rapid generation of new fibrils.

I recommend including PD brain material as seed (from patients), to allow elucidation of a more realistic situation. One must bear in mind that fibril polymorphism exists, and the seed characteristic may change the aggregation kinetic. There is a recent publication that explores this issue (<https://pubmed.ncbi.nlm.nih.gov/36802500/>).

To ensure that our conclusions also apply to the fibril polymorphs present in disease, we have now measured CSF samples from both healthy and Parkinson's disease patients, to investigate their ability to seed aggregation and oligomer formation. This now forms a new results section in the revised manuscript: "Parkinson's disease patient samples catalyse oligomer formation through secondary nucleation". In brief, we found that CSF from PD seeded aggregation significantly more effectively than healthy control CSF (also confirmed by RT-QuIC using our labelled alpha-synuclein construct), and that oligomers with the

same biophysical properties as in our in vitro seeded reactions were formed. This finding provides strong evidence for secondary nucleation as a source of oligomers in PD.

TEM is not a quantitative technique; thus, it is not straightforward to comment on fibril fragmentation based solely on the TEM images.

Reviewer 2 had a similar comment, our reply is copied here for convenience:

We agree that it is possible for TEM to distort the measured fibril length distributions, due to the deposition step as noted by the reviewer. However, as we are mainly concerned with the *changes* in length distribution over time, such biases, which will exist in all timepoints, will have a less pronounced effect. Nonetheless, we sought to provide an alternative measure to confirm the dominant mechanism was secondary nucleation not fragmentation. For length distribution measurements, TEM is considered a state-of-the-art method and other surface deposition-based methods such as atomic force microscopy and super-resolution microscopy also come with the same limitations. Solution-based methods such as dynamic light scattering, fluorescence correlation spectroscopy, and microfluidic diffusional sizing, are unsuitable for measuring fibril length distributions given the large size, non-spherical geometry and polydispersity of the aggregation samples.

Therefore, we have also now performed complementary experiments which support our conclusion that fragmentation is not the dominant mechanism by which new fibrils form which do not rely on measurements of the length distribution. We performed aggregation reactions in the presence of the Brichos chaperone domain, a well-characterised inhibitor of secondary nucleation, but not fragmentation. The aggregation was markedly slowed by Brichos, indicating that secondary nucleation, not fragmentation, is the main mechanism of fibril formation. These data are presented in figure S13 and lines 126-139 of the main manuscript text.

Authors comment that they use physiologically relevant conditions, but the assay medium is far from physiological. One must consider mimicking the crowded cellular environment.

We did not intend to claim that our reaction conditions are a perfect mimic of the cellular environment, merely that the conditions were chosen to more closely resemble the cytosol in terms of pH and salt concentration than some previous works. We have edited the manuscript to clarify this point, removing the phrase “physiologically relevant”..

More generally, while we do not expect that in vitro-determined rate constants are preserved in aggregation in the real cellular environment, a large body of work supports the notion that in vitro-determined mechanisms are indeed predictive of in vivo behaviour (Meisl and Xu et al. Science Advances (8), 2022).

It would be valuable to discuss how to address PD therapeutically based on the different aggregation processes (primary or secondary nucleation).

The implications of our work on therapeutic strategies are a key conclusion, and we have expanded our discussion of the results to reflect this, by the addition of the following sentence in lines 275-279: “Our finding that secondary nucleation is not only a pathway to the formation of new fibrils but also the main source of oligomeric species, highlights it as

a promising therapeutic target with dual effectiveness: stopping secondary nucleation will stop oligomer production in the short term and slow fibril accumulation in the long term.”

Reviewer #4 (Remarks to the Author):

Review of:

Rapid single molecule detection reveals that α -synuclein oligomers form by secondary nucleation

Xu et al.

Nice paper!

1. Key results; Clarity and Context; Significance

In the abstract, the authors wrote:

"Here we first show that, at physiological pH, α -synuclein aggregates by secondary nucleation, rather than fragmentation, and that this process is enhanced by agitation."

Conclusions of this reviewer:

- The perception (importance of secondary nucleation) is not new.
 - The use of single molecule detection is not a key, compelling new technology applied in such investigations and offering new insights. New variants of ultra-resolution microscopy, as one counterexample, could/would have been. In any event, one might wish to (have) emphasize(d) the methodology, perhaps in a submission to something like Nature Methods (?), along the lines "Rapid single molecule detection in microfluidic free-flow electrophoresis confirms that α -synuclein oligomers form by secondary nucleation".
- Our reply to a similar comment from reviewer 3: "Secondary nucleation has so far only been reported under acidic (pH 4.8 (Buell et al. PNAS (21), 2014) and 5.5 (Gaspar et al. Q Rev Biophys (50), 2017) conditions, and only for fibril formation. Here, we demonstrate that secondary nucleation not only takes place at neutral pH, but is also the major driver of oligomers, proposed as a key toxic species in Parkinson's disease. This work thus presents a large advance in our understanding of alpha-synuclein aggregation and provides a new consideration of therapeutic design. Previously, secondary nucleation of alpha-synuclein was only thought to be relevant in vivo upon entry into acidic lysosomal compartments. Our new findings demonstrate that secondary nucleation of alpha-synuclein can occur at neutral pH, and even on aggregates extracted from PD brains (new data) and may therefore be a ubiquitous and central process in pathology, contributing both to toxic oligomer production and rapid generation of new fibrils."

The authors do not cite a number of recent publications (examples: refs. r1-r13 below; the first 8 published in 2023) dealing with important aspects of the secondary nucleation of AS aggregation, i. e. key factors determining the course of reaction: modification status of the

AS monomer, solution conditions, temperature, agitation, small molecule reaction inhibitors or accelerators. The uncited references listed below include 7 papers (r1, r4, r5, r7, r8, r9, r12) co-authored by one or both of the senior authors of the present submission, M. Vendruscolo and T.P.J. Knowles. In fact, the titles (and abstracts) of r5 and the present submission offer the same observation regarding secondary nucleation. In addition, the authors have pre-published virtually the identical paper (bioRxiv), which apparently did not lead to significant changes, improvements. [This reviewer confesses to being a non-proponent of prepublication.]

We thank the reviewer for their extensive work into recent literature. We have added several new citations based on the reviewers' suggestions and note that r12 was already included as a reference.

r1. Staats, R., Brotzakis, Z.F., Chia, S., Horne, R.I., and Vendruscolo, M. (2023).

Optimization of a small molecule inhibitor of secondary nucleation in alpha-synuclein aggregation. *Front Mol Biosci* 10, 1155753. 10.3389/fmolb.2023.1155753.

r2. Pandit, E., Das, L., Das, A.K., Dolui, S., Saha, S., Pal, U., Mondal, A., Chowdhury, J., Biswas, S.C., and Maiti, N.C. (2023). Single point mutations at the S129 residue of alpha-synuclein and their effect on structure, aggregation, and neurotoxicity. *Front Chem* 11, 1145877. 10.3389/fchem.2023.1145877.

r3. Majid, N., Siddiqi, M.K., Hassan, M.N., Malik, S., Khan, S., and Khan, R.H. (2023). Inhibition of primary and secondary nucleation along with disruption of amyloid fibrils and alleviation of associated cytotoxicity: A biophysical insight of a novel property of Chlorpropamide (an anti-diabetic drug). *Biomater Adv* 151, 213450. 10.1016/j.bioadv.2023.213450.

r4. Horne, R.I., Murtada, M.H., Huo, D., Brotzakis, Z.F., Gregory, R.C., Possenti, A., Chia, S., and Vendruscolo, M. (2023). Exploration and Exploitation Approaches Based on Generative Machine Learning to Identify Potent Small Molecule Inhibitors of alpha-Synuclein Secondary Nucleation. *J Chem Theory Comput* 19, 4701-4710. 10.1021/acs.jctc.2c01303.

r5. Horne, R.I., Metrick, M.A., 2nd, Man, W., Rinauro, D.J., Brotzakis, Z.F., Chia, S., Meisl, G., and Vendruscolo, M. (2023). Secondary Processes Dominate the Quiescent, Spontaneous Aggregation of alpha-Synuclein at Physiological pH with Sodium Salts. *ACS Chem Neurosci* 14, 3125-3131. 10.1021/acschemneuro.3c00282.

r6. Frey, L., Ghosh, D., Qureshi, B.M., Rhyner, D., Guerrero-Ferreira, R., Pokharna, A., Kwiatkowski, W., Serdiuk, T., Picotti, P., Riek, R., and Greenwald, J. (2023). On the pH-dependence of α -synuclein amyloid polymorphism and the role of secondary nucleation in seed-based amyloid propagation. *bioRxiv* p. 10.1101/2023.06.25.546428.

r7. Dada, S.T., Hardenberg, M.C., Toprakcioglu, Z., Mrugalla, L.K., Cali, M.P., McKeon, M.O., Klimont, E., Michaels, T.C.T., Knowles, T.P.J., and Vendruscolo, M. (2023). Spontaneous nucleation and fast aggregate-dependent proliferation of alpha-synuclein aggregates within liquid condensates at neutral pH. *Proc Natl Acad Sci U S A* 120, e2208792120. 10.1073/pnas.2208792120.

r8. Bell, R., Castellana-Cruz, M., Nene, A., Thrush, R.J., Xu, C.K., Kumita, J.R., and Vendruscolo, M. (2023). Effects of N-terminal Acetylation on the Aggregation of Disease-related alpha-synuclein Variants. *J Mol Biol* 435, 167825. 10.1016/j.jmb.2022.167825.

- r9. Zurlo, E., Kumar, P., Meisl, G., Dear, A.J., Mondal, D., Claessens, M., Knowles, T.P.J., and Huber, M. (2021). In situ kinetic measurements of alpha-synuclein aggregation reveal large population of short-lived oligomers. PLoS One 16, e0245548. 10.1371/journal.pone.0245548.
- r10. Sakunthala, A., Datta, D., Navalkar, A., Gadhe, L., Kadu, P., Patel, K., Mehra, S., Kumar, R., Chatterjee, D., Sengupta, K., et al. (2021). Size-dependent secondary nucleation and amplification of α -synuclein amyloid fibrils. bioRxiv 10.1101/2021.12.28.474324.
- r11. Peduzzo, A., Linse, S., and Buell, A.K. (2020). The Properties of alpha-Synuclein Secondary Nuclei Are Dominated by the Solution Conditions Rather than the Seed Fibril Strain. ACS Chem Neurosci 11, 909-918. 10.1021/acscchemneuro.9b00594.
- r12. Tornquist, M., Michaels, T.C.T., Sanagavarapu, K., Yang, X., Meisl, G., Cohen, S.I.A., Knowles, T.P.J., and Linse, S. (2018). Secondary nucleation in amyloid formation. Chem Commun (Camb) 54, 8667-8684. 10.1039/c8cc02204f.
- r13. Roberti, M.J., Folling, J., Celej, M.S., Bossi, M., Jovin, T.M., and Jares-Erijman, E.A. (2012). Imaging nanometer-sized alpha-synuclein aggregates by superresolution fluorescence localization microscopy. Biophys J 102, 1598-1607. 10.1016/j.bpj.2012.03.010.

2. Results; Validity; Data and Methodology

- The authors wrote: ". Despite its heavy implication in Parkinson's disease, the aggregation of α -synuclein remains relatively poorly characterized."

Not true, there are few processes in biology that have received more attention.

It is true that alpha-synuclein has been heavily investigated. However, despite the large amount of attention, the molecular mechanisms of aS aggregation are much less understood than e.g. Abeta in Alzheimer's disease. We have edited the text to more clearly express our intended meaning, which now reads (lines 62-66) "Although alpha-synuclein oligomers are implicated as toxic species in PD, the molecular mechanisms by which both they and high molecular weight aggregates form remain largely unknown, despite substantial efforts. However, in order to enable the rational design of drugs that target aggregation, for example inhibiting the formation of toxic oligomeric species, this mechanistic information is required."

- α -Synuclein aggregation occurs via secondary pathways. This section is ok

- Secondary nucleation is the dominant mechanism of fibril formation.

There is a problem here, due to the lack of an evaluation/interpretation by the microscopy of Figs. 3, S9, S10, with respect to the incidence and length of secondary initiated fibrils. Such a visualization has been performed in the past, such as by super-resolution fluorescence microscopy (r13 above) in which secondary anchors were visualized and the rate of elongation (nm/h) of secondary fibrils was estimated as "7 \pm 4 nm/min". The authors of the present submission have actually published a mean (ensemble: primary+secondary) elongation rate (Supplementary of the seminal paper Buell et al., 2014: " At 37 ° C and 20 μ M soluble protein concentration, α -synuclein fibrils grow on average 1-2 nm/min in PBS

buffer, assuming a fibril diameter of 7 nm, and using the value of $k +$ determined here."). A careful consideration of these values based on comparison of conditions, etc. is beyond the scope of this review, yet considering that secondary fibrils may grow by different rates of radial as well as axial accretion of monomers, the direct visualization and evaluation of primary and secondary fibrils is essential. The methods (Fiji based) used on the TEM data of the present submission did not permit this nor do the authors address the importance of the elongation rate directly.

To clarify, fibrils formed by primary and secondary nucleation have not been distinguishable by any method to date, and are, for all we know, identical. They are all linear structures that grow from their ends, and this linear growth model forms the basis for all our kinetic models.

- fitting of oligomer dynamics

The submitting Cambridge lab has published extensively on kinetic models for amyloid aggregation. Nonetheless, this reviewer chooses to question features of Eqs. 4 and 5 used to evaluate the kinetic data of the present submission. Eq. 4 assumes a concerted (questionable?) interaction of $n+1$ monomers to form a progenitor fibril and also assumes a simple first order dissociation of monomers from fibrils. However, the indications are that monomers can dissociate along the length of a fibril as well as from the ends such that a proper representation would require an additional length dependent term. The same consideration applies to Eq. 5. It is neither the desire nor intention of this reviewer to engage in a protracted "kinetic" controversy but it may be expedient for the authors to reconsider some of the assumptions of their model(s). If some modifications are necessary/indicated, the quantitative data analysis and the conclusions of the paper need to be reexamined.

We thank the reviewer for their interest in our kinetic modelling. The focus of this work is on the new data, and the models we use herein were developed in previous publications. We refer the reviewer to these for a detailed discussion of the molecular origins of each of these terms in the equations, for example:

Dynamics of oligomer populations formed during the aggregation of Alzheimer's A β 42 peptide. Michaels et al. Nature Chemistry (12), 2020.

Identification of on- and off-pathway oligomers in amyloid fibril formation. Dear et al. Chemical Science (11), 2020.

- Oligomers form by secondary nucleation on fibril surfaces

The authors applied μ FFE to examine the oligomer dynamics: "A key feature of this approach in studying oligomers is its minimal perturbation of the reaction system. The sample under study is rapidly diluted and fractionated in solution just a few milliseconds prior to measurement, a timescale on which the sample composition is unlikely to change." Fig. S11 shows that oligomers have a higher mobility than monomers in the electric field. However, the regions selected in the figure seem somewhat arbitrarily placed and the zone between them represents a substantial amount of material. Also why is the monomer peak so broad?

We have now clarified this issue in the figure caption for S14 (previously S11) and in the methods section. The highlighted regions in figure S14 are for illustrative purposes, and in the oligomer detection analysis the whole chamber volume was considered, not only the indicated regions in S14, although this is indeed the region in which most oligomers are observed. The monomer peak is broadened upon application of the electric field due to the non-homogeneous flow profile resulting from interactions with the device surface. The flow rate is slower near the channel wall than in the centre of the channel cross-section, so that protein molecules located at different vertical heights through the channel experience the lateral electrophoretic force to a different extent relative to their movement in the flow direction, resulting in a broadening of the peak. This is something we have previously observed (Arter and Xu et al. Nano Letters (11), 2020), and to reduce this effect in this study we performed scans across the centre of the device cross-section to decrease the observed peak broadening. While this is now decreased in comparison to our previous work with widefield fluorescence imaging, the broadening effect is still present due to the high concentration and brightness of monomer and the non-digital nature of the laser spot.

Reviewers' Comments:

Reviewer #1:

Remarks to the Author:

the authors have gone a long way to addressing comments.

Reviewer #3:

Remarks to the Author:

I am pleased with the extensive work the authors have done to address all reviewers' comments, including mine. The inclusion of details regarding the methodology and data analysis, as well as the response to other reviewers, has significantly improved the clarity of the manuscript. In particular, the use of cerebrospinal fluid (CSF) samples from both healthy individuals and Parkinson's disease (PD) patients supports the occurrence of secondary nucleation in vivo, providing a more realistic sample condition and enabling a more thorough discussion regarding therapeutic interventions for PD. The discussion with the other reviewers was also very productive. I now agree that the work is robust and suitable for publication.